# BRP-NAS: Prediction-based NAS using GCNs

**Łukasz Dudziak[1*], Thomas Chau[1*], Mohamed S. Abdelfattah[1]**
**Royson Lee[1], Hyeji Kim[3†], Nicholas D. Lane[1,2]**

[1] Samsung AI Center, Cambridge, UK
[2] University of Cambridge, UK      [3] University of Texas at Austin, US
[*] *Indicates equal contributions*, [†] *Work performed while at Samsung AI Center*

{l.dudziak, thomas.chau, mohamed1.a, royson.lee, nic.lane}@samsung.com
hyeji.kim@austin.utexas.edu

## Abstract

Neural architecture search (NAS) enables researchers to automatically explore broad design spaces in order to improve efficiency of neural networks. This efficiency is especially important in the case of on-device deployment, where improvements in accuracy should be balanced out with computational demands of a model. In practice, performance metrics of model are computationally expensive to obtain. Previous work uses a proxy (e.g., number of operations) or a layer-wise measurement of neural network layers to estimate end-to-end hardware performance but the imprecise prediction diminishes the quality of NAS. To address this problem, we propose BRP-NAS, an efficient hardware-aware NAS enabled by an accurate performance predictor-based on graph convolutional network (GCN). What is more, we investigate prediction quality on different metrics and show that sample efficiency of the predictor-based NAS can be improved by considering binary relations of models and an iterative data selection strategy. We show that our proposed method outperforms all prior methods on NAS-Bench-101 and NAS-Bench-201, and that our predictor can consistently learn to extract useful features from the DARTS search space, improving upon the second-order baseline. Finally, to raise awareness of the fact that accurate latency estimation is not a trivial task, we release LatBench – a latency dataset of NAS-Bench-201 models running on a broad range of devices.

## 1 Introduction

Neural architecture search (NAS) has demonstrated great success in automatically designing competitive neural networks compared with hand-crafted alternatives [1, 2, 3, 4]. However, NAS is computationally expensive requiring to train models [5, 6] or introduce non-trivial complexity into the search process [7, 8]. Additionally, real-world deployment demands models meeting efficiency or hardware constraints (e.g., latency, memory and energy consumption) on top of being accurate, but acquiring various performance metrics of a model can also be time consuming, independently from the cost of training it.

In this paper, we (*a*) show the limitations of the layer-wise predictor both in terms of prediction accuracy and NAS performance and (*b*) propose a *Graph convolutional networks* (GCN)-based predictor for the end-to-end latency which significantly outperforms the layer-wise approach on devices of various specifications.

One of the key challenges in obtaining a reliable accuracy predictor is that acquiring training samples (pairs of *(model, accuracy)*) is computationally expensive. *Sample efficiency* denotes how many samples are required to find the best model during a search under a target hardware constraint, and

significantly advancing this metric is a key contribution of our work. We propose several methods to improve sample efficiency – ($a$) We observe that in the context of NAS, instead of getting precise estimates of accuracy, we want to produce a linear ordering of accuracy to search for the best model. Therefore, we propose a binary relation predictor to decide the accuracy ranking of neural networks without requiring to estimate absolute accuracy values. ($b$) To help the predictor focus on predicting the rankings of top candidates, which is the most important to yield the best results in NAS, we propose an iterative data selection scheme which vastly improves the sample efficiency of NAS.

The contributions of this paper are summarized as follows:

- **Latency prediction.** We empirically show that an accurate latency predictor plays an important role in NAS where latency on the target hardware is of interest, and existing latency predictors are overly error-prone. We propose an end-to-end NAS latency predictor based on a GCN and show that it outperforms previous approaches (proxy, layer-wise) on various devices. To the best of our knowledge, this is the first end-to-end latency predictor. We illustrate its behaviour on various devices and show that this predictor works well across all of them (Section 3).

- **Accuracy prediction.** We introduce a novel training methodology for a NAS-specific accuracy predictor by turning an accuracy prediction problem into a binary prediction problem, where we predict which one of two neural architectures performs better, resulting in improved overall ranking correlation between predicted and ground-truth rankings. Furthermore, we propose a new prediction-based NAS framework called BRP-NAS. It combines a binary relation accuracy predictor architecture and an iterative data selection strategy to improve the top-K ranking correlation. BRP-NAS outperforms previous NAS methods by being more sample efficient (Section 4 and 5).

- **Towards reproducible research: latency benchmark.** We introduce *LatBench*, the first large-scale latency measurement dataset for multi-objective NAS. Unlike existing datasets which either approximate the latency or focus on a single device, LatBench provides measurement dataset on a broad range of systems covering desktop CPU/GPU, embedded GPU/TPU and mobile GPU/DSP. We also release *Eagle* which is a tool to measure and predict performance of models on various systems. We make LatBench and the source code of Eagle available publicly[1] (Section 6).

## 2    Related work

**NAS and performance estimation.** Various end-to-end predictors have been proposed and studied for accuracy. Initially, accuracy predictors were shown to be helpful in guiding a NAS [9]. More recently, it has been shown that accuracy predictor alone can be used to perform a search [10, 11].

Performance of a neural network is captured by several metrics, such as accuracy, latency, and energy consumption. Because measuring performance metrics is expensive both in terms of time and computation, interest in predicting them has surged since neural architecture search was introduced in [5]. Among several metrics, accuracy prediction is arguably the most actively studied in the context of NAS. Earlier performance prediction works focused on extrapolating the learning curve to reduce the training time [12, 13, 14]. Recent works, on the other hand, explored performance prediction based on architectural properties. For instance, it was demonstrated that an accuracy predictor trained during the search could be successfully used to guide it and, in turn, accelerate it [9]. Taking advantage of the differentiability of accuracy predictors, NAO [15] introduced a gradient-based optimization of neural architectures. Graph neural networks, including GIN [16], D-VAE [17] and GATE [18] have been used to learn representations of neural architectures. In this paper, we use GCN [19] to handle neural architectures described as directed graphs. Closely related to our work, NPENAS [10] used graph neural network-based accuracy predictors and an iterative approach to estimate the accuracy of models. However, instead of training the predictor with the top-k mutated models based on previously found models, we trained the predictor by picking both the top-k and random models from the entire search space, an technique that balances exploration and exploitation. Additionally, we incorporate transfer learning to further boost the performance of the predictor as shown in Section 4.1.

Recent focus has been on improving sample efficiency. In that regard, [20] proposed adapting the action space during NAS, where a Monte-Carlo tree search was used to split the action space into good and bad regions. Towards the same goal, [11] introduced a simple NAS based on accuracy predictor, where models with the top-K best predicted accuracy were fully trained, after which the best one was chosen. Similarly, BONAS [21] incorporated a GCN accuracy predictor as a surrogate function of Bayesian Optimization to perform NAS and used exponential weighted loss to improve the prediction of models with high accuracy. Our work extends GCN to binary relation learning to focus on the prediction of ranking rather than absolute values and uses iterative data selection to explicitly train the predictor with high performing models.

**Focused latency prediction.** Several works have been proposed to predict performance metrics such as accuracy and latency, the two most popular metrics of interest, instead of measuring them [9, 10, 11]. Recent work either measures the latency directly from devices during the search process [22, 23], which is accurate but slow and expensive, or rely on a proxy metric (e.g., FLOPS or model size) [24], which is fast but inaccurate. More recently, layer-wise predictors have been proposed [4] which effectively sum up the latencies of individual neural network layers. While being more accurate than the proxy, layer-wise predictors have a significant drawback: they do not capture the complexities of multiple layer executing on real hardware.

**Multiobjective NAS.** A few works have explored NAS with multiple objectives and hardware constraints. Among those, [25] proposed a hardware-aware adaptation of neural architectures via evolutionary search, where the performance metrics of each architecture were estimated by the predictors. Latency was estimated via a lookup table while accuracy and energy consumption predictors were modeled as Gaussian process regression.

# 3 Latency prediction in NAS

In this section, we demonstrate the limitations of existing latency predictors and introduce a GCN-based latency predictor which ($a$) significantly outperforms the existing predictors on a wide range of devices in absolute accuracy and ($b$) contributes to a significant improvement in NAS for latency constrained deployment. Throughout the paper, we focus on NAS-Bench-201 dataset which includes 15,625 models. We use desktop CPU, desktop GPU and embedded GPU to refer to the devices used in our analysis, with device details described in Section 6.

## 3.1 Existing latency predictors and their limitations in NAS

Number of FLOPS and parameters are often used as proxies for latency estimation for their simplicity but have been shown to be inaccurate in many cases [23, 26]. In Figure 1 (left), we show the scatter plot of models taken from NAS-Bench-201 dataset that illustrates the connection between the latency and FLOPS. Each point in the plots represents the average latency of running a model on the stated device. We can see that latency is not strongly correlated with FLOPS. Recent works [4, 23] use a layer-wise predictor which derives the latency by summing latency measured for each operation in the model individually. However, as shown in Figure 1 (middle), the layer-wise predictor[2] also leads to inaccurate predictions of the end-to-end latency. It assumes sequential processing of operations and cannot reflect the key model and hardware characteristics that affect the end-to-end latency, e.g., whether operations within a model can be executed in parallel on the target hardware. In section 3.2, we introduce an end-to-end latency predictor that is trained with the end-to-end measured latency that significantly improves the prediction accuracy as shown in Figure 1 (right).

**NAS with latency predictors.** We analyze the impact of layer-wise latency predictor on NAS for latency-constrained deployment, where the objective is to find the most accurate model that satisfies a strict latency constraint (e.g., for real-time applications [27, 28]). We consider two NAS algorithms, oracle NAS and Aging Evolution [29] (AE). Oracle NAS can find the best performing model (in terms of accuracy) among those with latency satisfying the target constraint – therefore its performance is limited entirely by the quality of latency estimation (more details are provided in the S.M.). In Figure 2 (left), we plot the difference between the best achievable accuracy and the best accuracy on CIFAR-100 dataset obtained by an oracle NAS that relies on the predicted latency - as a function of the target latency constraint. We can see that the accuracy loss due to the inaccurate predictor is non-negligible and sometimes very large (up to $8\%$). This loss is also visible when other NAS algorithms are used. In Figure 2 (right), we plot the best accuracy of models found by the aging-evolution search

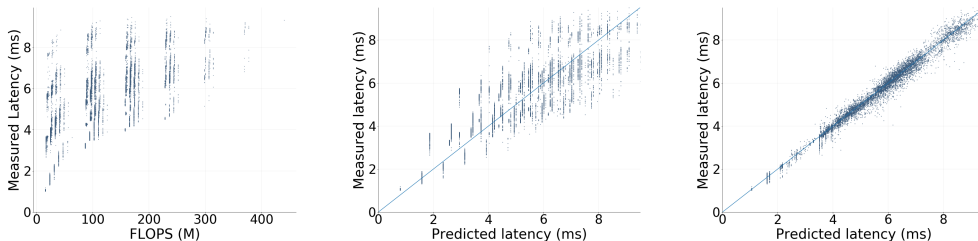

Figure 1: (Left) Number of FLOPS is not a good proxy for the latency estimation. Our GCN-based end-to-end latency predictor (right) is more accurate than layer-wise predictor (middle). Results shown here are based on desktop CPU.

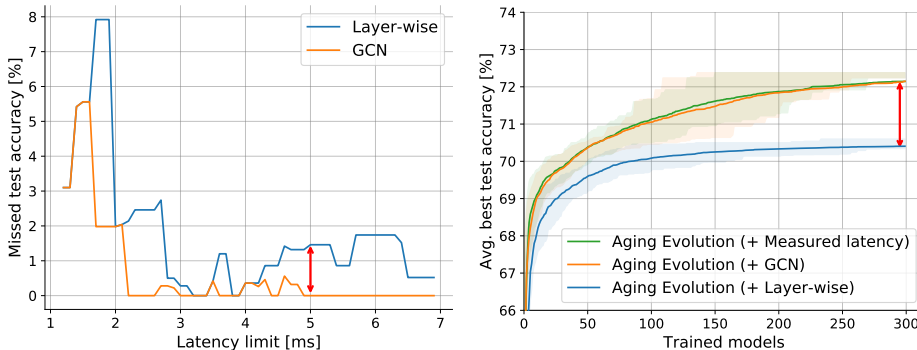

Figure 2: Importance of an accurate latency predictor in NAS for latency-constrained deployment – (Left) Best achievable models missed when using an oracle NAS that relies on predicted latency on desktop GPU. The gap between the layer-wise and GCN curves shows the impact of poor latency predictor. (Right) Best accuracy of models found by the aging-evolution search with predicted latency and measured latency on desktop GPU with 5ms latency limit. Shaded regions mark interquartile range and the same applies to subsequent figures.

with predicted latency and measured latency as a function of search step. These experimental results highlight the importance of an accurate latency predictor in NAS.

**Analysis of Pareto-optimal models.** In order to systematically study the impact of inaccurate predictions on latency-aware NAS, we run an analysis of the Pareto-optimal models. Pareto-optimal models are solutions to NAS given a strict latency constraint or if the objective is a weighted combination of the accuracy and the latency [30]. We ask the following: can the Pareto-optimal models be discovered when a latency predictor is used? Suppose we are given an oracle NAS algorithm that returns a Pareto-optimal model based on the accuracy and *predicted* latency. How far off is that model from the desired Pareto-optimal solution in the accuracy and *measured* latency plot? In Figure 3 (left and middle), we show scatter plots of NAS-Bench-201 [31] models, where the y-axis represents the accuracy and the x-axis represents latency predicted via the layer-wise predictor (left), and measured latency (middle), respectively. Pareto-optimal models in the predicted space and measured space are marked with pink (o) and red (x), respectively, and shown in both figures. We can see that the Pareto-optimal models in one space do not always lie at the Pareto frontier of the other space. This is problematic because it implies that even if we had a perfect NAS algorithm that discovers Pareto optimal points (based on the predicted latency), there is a high chance that the discovered models would not be Pareto optimal in practice.

## 3.2 End-to-end GCN-based latency predictor.

Our proposed end-to-end latency predictor consists of a GCN which learns models for graph-structure data [19]. Given a graph $g = (V, E)$, where $V$ is a set of $N$ nodes with $D$ features and $E$ is a set of edges, a GCN takes as input a feature description $X \in \mathbb{R}^{N \times D}$ and a description of the graph structure as an adjacency matrix $A \in \mathbb{R}^{N \times N}$. For an $L$-layer GCN, the layer-wise propagation rule is the following:

$$H^{l+1} = f(H^l, A) = \sigma\left(AH^lW^l\right),$$

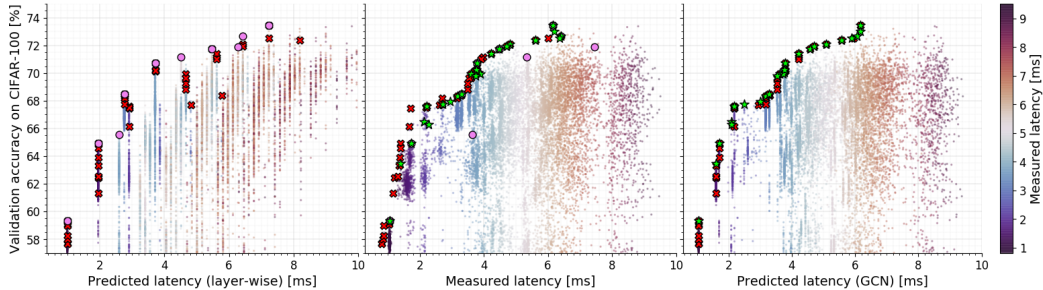

Figure 3: Red (x), Pink (o), Green marks (*) represent Pareto-optimal models on accuracy vs measured latency, layer-wise predicted latency, and GCN predicted latency, respectively, on desktop GPU. (Left) Many Pareto-optimal models (red, x) are not located at the Pareto-frontier implying that an oracle NAS cannot discover Pareto-optimal models with the layer-wise predicted latency. (Right) Most Pareto-optimal points (red, x) are located at the Pareto-frontier implying that an oracle NAS is able to discover Pareto-optimal models with our GCN predicted latency.

where $H^l$ and $W^l$ are the feature map and weight matrix at the $l$-th layer respectively, and $\sigma(\bullet)$ is a non-linear activation function like ReLU. $H^0 = X$ and $H^L$ is the output with node-level representations.

**Architecture.** Our GCN predictor has 4 layers of GCNs, with 600 hidden units in each layer, followed by a fully connected layer that generates a scalar prediction of the latency. The input neural network model to the GCN is encoded by an adjacency matrix A (asymmetric as the computation flow is represented as a directed graph) and a feature matrix X (one-hot encoding). We also introduce a global node (the node that connects to all the other node) to capture the graph embedding of neural architecture by aggregating all node-level information. GCN can handle any set of neural network models. The details of models used in this paper are in the S.M.

**Training.** All predictors are trained for 100 times, each time using a randomly sampled set of 900 models from the NAS-Bench-201 dataset. 100 random models are used for validation and the remaining 14k models are used for testing.

**Results.** Our GCN predictor outperforms existing predictors, establishing new state-of-the-art (Figure 1), and demonstrates strong performance suitable for NAS (Figure 3). In Table 1, we show the performance of the proposed GCN latency predictor comparing to the layer-wise predictor on various devices. The values are the percentage of models with predicted latency within the corresponding error bound relative to the measured latency. We can see that the strong performance generalizes across various devices, which have vastly different latency behaviors. We provide an extensive study on the latency behavior on various devices in the S.M.

Table 1: Performance of latency predictors on NAS-Bench-201: Our GCN predictor demonstrates significant improvement over the layer-wise predictor across devices. More results in the S.M.

| Error bound | Accuracy of GCN predictor [%] | | | Accuracy of Layer-wise predictor [%] | | |
|---|---|---|---|---|---|---|
| | Desktop CPU | Desktop GPU | Embedded GPU | Desktop CPU | Desktop GPU | Embedded GPU |
| ±1% | 36.0±3.5 | 36.7±4.0 | 24.3±1.4 | 3.5±0.2 | 4.2±0.2 | 6.1±0.3 |
| ±5% | 85.2±1.8 | 85.9±1.9 | 82.5±1.5 | 18.2±0.4 | 17.1±0.3 | 29.7±0.8 |
| ±10% | 96.4±0.7 | 96.9±0.8 | 96.3±0.5 | 29.6±1.1 | 32.6±1.2 | 54.0±0.8 |

## 4   Accuracy prediction in NAS

In the previous section, we assumed that the accuracy of the model is freely available during the search and focused on the latency prediction. In practice, accuracy of the model is computationally expensive to obtain, sometimes more than latency, as it requires training. The cost of NAS critically depends on the sample efficiency, which reflects how many models need to be trained and evaluated during the search.

In this section, we ($a$) propose a new prediction-based NAS framework, called Binary Relation Predictor-based NAS (*BRP-NAS* in short), that combines a GCN binary relation predictor and a novel

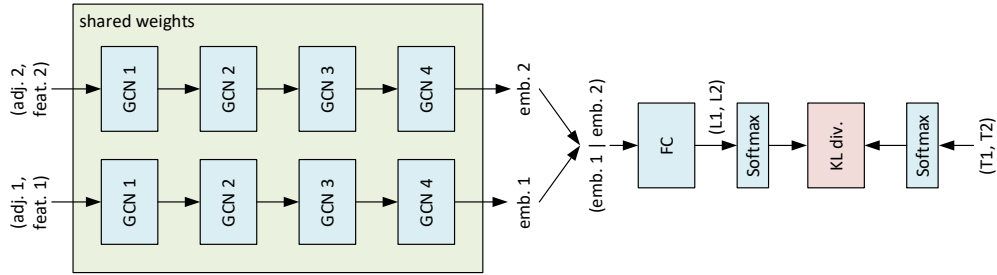

Figure 4: Overview of the proposed approach to train a binary relation predictor.

iterative data selection strategy; (*b*) demonstrate that it vastly improves the sample efficiency of NAS for accuracy optimization.

## 4.1 Transfer learning from latency predictors to improve accuracy predictors

We demonstrate that latency predictors are surprisingly helpful in improving the training of an accuracy predictor – both in terms of absolute accuracy and the resulting NAS performance. The idea is that these GCN-based predictors for latency, accuracy and FLOPS have the same input representation. A trained GCN (e.g., for latency prediction) captures features of a model which are also useful for a similar GCN trained to predict a different metric (e.g., accuracy).

To show that, we initialize the weights of an accuracy predictor with those from a latency/FLOPS predictor. We then train the predictor using the validation accuracy of CIFAR-100 dataset. All predictors are trained for 100 times, each time using a randomly sampled set of 100 models from NAS-Bench-201. Another 100 random models are used for validation and the remaining 14k models are left for testing. Table 2 shows that the quality of accuracy prediction improves in all cases, in particular, the accuracy predictors with transfer learning from FLOPS, which is freely available, can increase the sample efficiency by around 2 times. The proposed transfer learning method is applicable to any accuracy predictors with existing training techniques. We refer to the S.M. for more analysis.

Table 2: Performance of accuracy predictors on the test set. Standard refers to training with random initialization of the weights in GCN. Init-GPU and Init-FLOPS have the weights initialized with those of the desktop GPU latency predictor and FLOPS predictor, respectively. Transfer learning increases the sample efficiency by around 2 times.

| Error bound | GCN accuracy [%] | | | | | | |
|---|---|---|---|---|---|---|---|
| | 50 samples | | | 100 samples | | | 200 samples |
| | Standard | Init-GPU | Init-FLOPS | Standard | Init-GPU | Init-FLOPS | Standard |
| ±1% | 22.1±3.3 | 26.3±3.8 | 25.3±4.1 | 27.5±3.9 | 32.0±4.1 | 32.3±3.8 | 34.6±2.9 |
| ±5% | 72.7±3.0 | 74.8±3.4 | 73.7±3.6 | 76.9±2.4 | 80.5±2.2 | 80.5±2.7 | 81.7±1.8 |
| ±10% | 85.4±2.4 | 87.0±2.7 | 87.2±2.4 | 88.2±1.7 | 90.4±1.6 | 91.0±2.0 | 90.8±1.3 |

## 4.2 Binary relation predictor-based NAS

We propose a new predictor-based NAS according to the following observations: (*a*) accuracy prediction is not necessarily required to produce faithful estimates (in the absolute sense) as long as the predicted accuracy preserves the ranking of the models; (*b*) any antisymmetric, transitive and connex *binary relation* produces a linear ordering of its domain, implying that NAS could be solved by learning binary relations, where $O(n^2)$ training samples can be used from $n$ measurements; (*c*) accurately predicting the rankings of top candidates is the most important. (We refer to the S.M. for a more formal discussion on these observations and intuition behind them.)

BRP-NAS consists of two phases. In the first phase, the ranking of all candidate models is predicted based on the outputs from a binary relation predictor, which is trained to predict the binary relation (accuracy comparisons between two models). In the second phase, based on the predicted rankings,

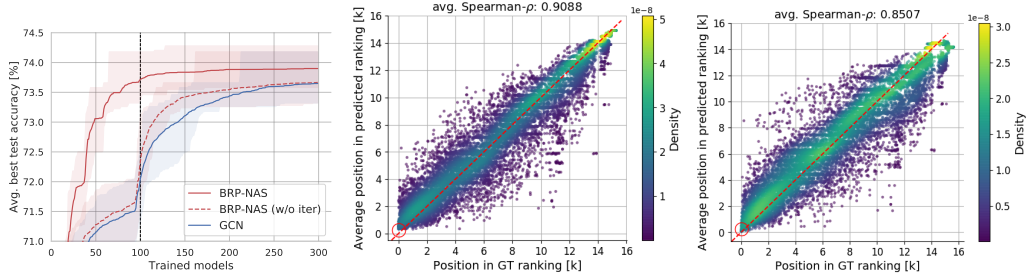

Figure 5: (Left) Binary relation prediction and iterative training significantly improve the performance of BRP-NAS over vanilla GCN approach. The dotted vertical line represents the point up to which the prediction was trained. (Middle) Ranking produced by the proposed binary relation predictor without iterative training data selection. (Right) Binary relation predictor trained via iterative data selection. As indicated by the red circle in the lower-left corners, even though the plain binary predictor achieves higher overall ranking correlation with respect to the ground-truth ranking, its accuracy comes from the better fit in the lower ranking models. Iterative data selection mitigates this issue, at the expense of the global ranking quality, by focusing its training on high performing models.

models with high *predicted* ranks are fully trained, after which, the model with the highest *trained* accuracy is selected.

**Binary relation predictor.** We propose a GCN based approach to learn a binary relation for NAS, as illustrated in Figure 4. The predictor reuses the GCN part of the latency predictor, without any changes, to generate graph embeddings for both input models. The embeddings are then concatenated and passed to a fully connected layer which produces a 2-valued vector. The vector is then passed through a softmax function to construct a simple probability distribution $p = (p_1, p_2) \in \mathbb{R}^2$ with $p_1$ being the probability of the first model *better* than the second, and $p_2$ being the probability of the opposite. The produced probability distribution is then compared to the target probability distribution obtained by taking a softmax of the ground-truth accuracy of the two models $(T1, T2)$, and the objective is to minimize the KL divergence between the two distributions. The overall network structure and the loss function are summarized in Figure 4.

**Training via iterative data selection.** Given a budget of $T$ models and $I$ iterations, we start by randomly sampling and training $T/I$ models from the search space, and the results are used to train the initial version of the predictor. At the beginning of each subsequent iteration, we use the predictor to estimate the accuracy of all the models, denoted by $M$, in the search space. We then select the top $\alpha * T/I$ unique models and randomly pick another $(1 - \alpha) * T/I$ models from the top $M/2^i$ models where $\alpha$ is a factor between 0 and 1 and $i$ is the iteration counter. The selected $T/I$ models are trained and their resulting accuracies are used to further train the predictor for the next iteration. Tuning $\alpha$ results in a trade-off between exploitation and exploration and we use $\alpha = 0.5$ for all our experiments.

**Results.** Figure 5 shows the advantages of the proposed binary relation and iterative data selection in BRP-NAS. Specifically, although the correlation between the measured ranking and the ground truth (GT) ranking decreases with iterative data selection (middle vs right), BRP-NAS is able to find better models because it focuses on high performing models only. This greediness, together with the increased sample efficiency from using binary relations, makes our BRP-NAS significantly better than the other considered predictors. More results (including ablation studies) can be found in the following section and in the S.M.

## 5 End-to-end results

**Comparison to prior work on NAS-Bench-201.** Figure 6 (left) shows results in an unconstrained setting where we compare to AE [29], REINFORCE [5], and random sampling. BRP-NAS outperforms other methods by being more than $2x$ more sample efficient than the runner up (AE). However, we also see that eventually the average performance of AE surpasses BRP-NAS – we investigated that and observed that this difference comes from the SGD-induced randomness. Specifically, unlike ours algorithm which trains each model only once, AE can (and often does) train the same model multiple times, which allows it to find good seeds more robustly. Please note that this only happens after sufficiently many models have been trained and in the presence of high SGD noise, therefore

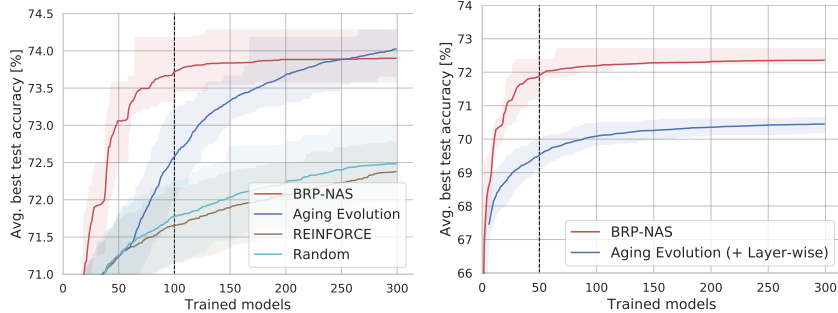

Figure 6: BRP-NAS outperforms Aging Evolution and other popular search methodologies on NAS-Bench-201, in a setting without latency constraints (left), and with a 5ms latency constraint on a desktop GPU (right).

constituting an edge case – for example, our method still consistently outperforms AE on a larger NAS-Bench-101 and in a constrained setting. More information about SGD-induced randomness can be found in the S.M.

We then combined BRP-NAS together with our GCN-based latency predictor from Section 3 to further improve NAS in the constrained settings. Similar to Figure 2, we compare to AE with a layer-wise predictor (current SOTA for latency prediction) – the results, shown in Figure 6 (right), demonstrate that the naive combination is far from optimal.

**Comparison to prior work on NAS-Bench-101.** To gauge the efficiency and quality of our BRP-NAS, we compare to previously published work (unconstrained case only). Simultaneously, this comparison shows how our technique scales to larger benchmarks as NAS-Bench-101 [32] contains over 423k points. Table 3 shows the result of this comparison, both in terms of the total number of trained models (algorithm speed) and final test accuracy on CIFAR-10 dataset (algorithm quality). We train the BRP-NAS predictor using the validation accuracy from 100 models, then we train the top 40 models returned by the predictor. With a total of 140 trained models, we are able to find a more accurate model faster than any previous work, as highlighted in Table 3 (detailed results in the S.M.).

Table 3: Comparison to prior work on NAS-Bench-101 dataset. Our result is averaged from 32 runs.

|                | NAO [15] | BONAS [21] | AE [29] | Wen et al. [11] | NPENAS [10] | BRP-NAS |
|----------------|----------|------------|---------|------------------|-------------|---------|
| Trained models | 1000     | 1000       | 418     | 256              | 150         | **140** |
| Test Acc. [%]  | 93.81    | **94.22**  | **94.22** | 94.17          | 94.14       | **94.22** |

**Results on the DARTS search space.** We further test our binary accuracy predictor on a much larger DARTS search space [33], performing classification on CIFAR-10. The first challenge we had to face was related to its size – with approximately $10^{18}$ models, it is infeasible to simply sort all of them. Because of that, we decided to begin a search by randomly selecting a subset of 10 million architectures and then run everything else analogically to the previous scenarios (20 models trained in each iteration, up to 60 models trained in total). We repeated the entire process 3 times, results are presented in Table 4. It is worth noticing that for a small search budget (6 GPU days), our BRP-NAS is in practice analogical to random search (since models trained in the first iteration are random). This suggests that the difference observed between different approaches in the "small budget" group is actually noise introduced by the random nature of algorithms. However, we can see that as we increase the number of trained models, our approach keeps improving results consistently, suggesting that the predictor is able to learn useful features and can robustly outperform DARTS on its search space. Further details and comments are provided in the S.M.

Table 4: Results on the DARTS search space for CIFAR-10 classification with different budgets.

|                   | Small budget |           |           | Medium budget |           | High budget |
|-------------------|--------------|-----------|-----------|---------------|-----------|-------------|
|                   | DARTS        | Random    | BRP-NAS   | Random        | BRP-NAS   | BRP-NAS     |
| Cost (GPU days)   | 6            | 6         | 6         | 30            | 30        | 60          |
| Test error [%]    | 2.76±0.09    | 2.87±0.06 | 2.71±0.07 | 2.75±0.18     | 2.66±0.09 | 2.59±0.11   |

# 6 Latency prediction benchmark

In this section, we present LatBench – a latency dataset of NAS-Bench-201 models on a wide range of devices. Similar to the motivations of NAS-Bench-101 and NAS-Bench-201, we aim towards $(a)$ reproducibility and comparability in hardware-aware NAS and $(b)$ ameliorating the need for researchers to have access to a broad range of devices. Although Nas-Bench-201 provides computational metrics such as number of parameters, FLOPS, and latency, these metrics are computed with operations and skip connections that do not contribute to the resulting output, leading to inaccurate measurements. Additionally, as latencies among devices often have weak correlations, more devices are required to facilitate the research of hardware-aware NAS.

We first remove any dangling nodes and edges and run each model in NAS-Bench-201 on the following devices: *(i)* Desktop CPU - Intel Core i7-7820X, *(ii)* Desktop GPU - NVIDIA GTX 1080 Ti, *(iii)* Embedded GPU - NVIDIA Jetson Nano, *(iv)* Embedded TPU - Google EdgeTPU, *(v)* Mobile GPU - Qualcomm Adreno 612 GPU, *(vi)* Mobile DSP - Qualcomm Hexagon 690 DSP.

Specifically, we run each model $1000$ times on each aforementioned non-mobile device using a patch size of $32 \times 32$ and a batch size of $1$. For mobile devices, each model is run $10$ time with the same settings. In order to lessen the impact of any startup/cool-down effects such as the creation and loading of inputs into buffer, we discard latencies that fall outside the lower and higher quartile values before taking the average of every 10 runs. These averages are discarded again with the aforementioned thresholds before a final average is taken.

For more details and further analysis of LatBench, please refer to S.M. We also provide the FLOPS and the number of parameters for these models after removing unneeded nodes and edges. We have plans of updating LatBench by adding more devices in the future.

# 7 Discussion and limitations

**Generalization to other metrics and problems.** Even though we focused our work on finding the most accurate architecture under a strict latency limit, we expect that our observations are still relevant under different settings. Generally speaking, our searching procedure can be described as an efficient predictor-based approach to solving multi-objective optimization with $\epsilon$-constraint method, where the emphasis is put on: *(a)* distinguishing between the constraining metric (latency in our case) and the optimised metric (accuracy), and *(b)* using different variations of predictors for them (unary and binary, respectively). Consequently, this approach can be used for other multi-objective optimisation problems. Furthermore, without distancing from NAS, the latency prediction part can naturally be extended to other hardware-related metrics.

**Sensitivity to design choices.** When designing our GCN predictor, we have tried different design choices – forward propagation (with a global sink) vs backward propagation (with a global source), and softmax vs sigmoid activations. They all lead to similar performance. We also notice that the performance of predictor is better without normalizing the adjacency matrix.

**Symmetry and transitivity of the learned relation.** Even though we motivate usage of the binary predictor by the fact that antisymmetric and transitive binary relations produce linearly ordered sets, we do not directly enforce any of these properties. Antisymmetry is enforced indirectly by including both orderings of pairs – $(m_1, m_2)$ and $(m_2, m_1)$ – in the predictor's training set. We empirically checked that on a randomly sampled set of 1000 models from the NAS-Bench-201 space (i.e., ~500k pairs) 98% of cases remain antisymmetric. To check transitivity, we investigated the number of simple cycles in a relation matrix for the same 1000 random models and found out that this number is exceptionally large (we stopped the program after reaching 10 million cycles). This suggests that, in general, our predictor does not guarantee transitivity, despite producing strong empirical results. We consider investigating the impact of cycles on NAS to be an interesting direction for future work.

# 8 Conclusion

We introduced BRP-NAS, a new prediction-based NAS framework that combines a binary relation accuracy predictor architecture and an iterative data selection strategy to improve the performance of NAS. BRP-NAS outperforms previous NAS methods in both sample efficiency and accuracy for NAS-Bench-101 and NAS-Bench-201, and also surpasses DARTS in its search space. We release LatBench – a latency dataset for models in the NAS-Bench-201, and Eagle – a tool to measure and predict performance of models on different devices.

## Broader Impact

This research can democratize on-device deployment with cost-efficient NAS methodology for model optimization within device latency constraints. Additionally, carbon footprint of traditionally expensive NAS methods is vastly reduced. On the other hand, measurement and benchmarking data can be used both to create new NAS methodologies, and to gain further insights about the device performance. This can bridge the machine learning and device research communities together.

## Funding Disclosure

This work was done as a part of the authors' jobs at Samsung AI Center. The authors have nothing to disclose.

## Footnotes

[1] https://github.com/thomasccp/eagle

[2]The layer-wise predictor is calibrated by a scaling factor to fit the end-to-end latency in the training set.

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
