[Supplementary Material · supplementary.pdf]

# Supplementary Material

## S1 Supplementary Material for Section 3: Latency prediction in NAS

### S1.1 Neural network models supported by GCN predictor

GCN can handle any set of neural network models. In this paper, we apply GCN to NAS-Bench-201 and NAS-Bench-101. Their network structures are described below.

In NAS-Bench-201, the skeleton of any model consists of 3 stacks of 5 cells with a fixed structure and placeholders for 6 operation nodes. In the original paper, this cell structure is described with the help of a directed acyclic graph whose nodes and edges represent tensors and data dependencies between them, respectively. Additionally, each edge is also assigned a label which defines the operation to apply to the source tensor and whose result is used to define the content of the destination tensor. Since the structure of the cell is fixed, the only "searchable" part of the cell are labels to be assigned to the edges – the authors considered 5 different options for each label: "zero" operation, "identity" operation (a.k.a. skip-connection), convolution $3 \times 3$, convolution $1 \times 1$, and $3 \times 3$ average pooling. Therefore, each architecture in NAS-Bench-201 can be defined by selecting 6 elements (with repetitions) from the aforementioned set of operations ( $O_i$ for $i = 1...6$) and represented with an architecture string: $|O_1 \sim 0| + |O_2 \sim 0| O_3 \sim 1| + |O_4 \sim 0| O_5 \sim 1| O_6 \sim 2|$ (as defined by the NAS-Bench-201 authors).

For the purpose of this work, we have modified the representation of the models in NAS-Bench-201 dataset in the following way:

- When constructing the graph representation of a network to use it with our GCN predictors, we begin by converting the NAS-Bench-201 cell graph (Figure S1 left) into its equivalent form using more traditional convention where nodes represent operations[3] (Figure S1 middle);

- We optimize the graph by completely detaching "zero" and "skip-connect"[4] operations, and then removing all other nodes which became dangling (i.e., they do not lay on the path from input to output) because of the previous step;

- As mentioned in Section 3.2, we add a global node which is connected to all other nodes (including the nodes which were detached due to optimizations) and also add self-connections for all nodes – this results in an adjacency matrix (Figure S1 right) with dimensions $9 \times 9$ (6 operation nodes, "input", "output" and "global" node) which is one of the inputs to the GCN;

- Finally, for each node we construct its feature vector by encoding the node's type using a one-hot vector – because "zero" and "skip-connect" operations were optimized out[5], the possible choices are: the three remaining operations plus "input", "output" and "global" node types – thus the feature matrix is $9 \times 6$.

In NAS-Bench-101, modules are represented by directed acyclic graphs with up to 7 nodes. The valid operations at each node are convolution $3 \times 3$, convolution $1 \times 1$, and $3 \times 3$ max pooling. This results in an adjacency matrix with dimensions $8 \times 8$ (5 operation nodes, "input", "output" and "global" node) and a feature matrix with dimensions $8 \times 6$ (3 operations plus "input", "output" and "global" node types).

### S1.2 Latency predictor for various devices

Table S2 and Figure S2 show the performance of the proposed GCN predictor. We first train each predictor for 100 times, each time using the hyperparameters summarized in Table S1 and a randomly sampled set of 900 models in the NAS-Bench-201 dataset. 100 models are used for validation and the remaining 14k models are used for testing. Values reported in Table S2 are the percentage of models in the test set that the predicted latency is within the corresponding error bound of the measured latency. The GCN predictors generalize well to unseen models in the NAS-Bench-201 dataset, and significantly outperform layer-wise predictors. We also see that the strong performance generalizes across various devices, which have vastly different latency behaviors. Then we experiment with training the GCN predictors using a smaller randomly sampled set of 100 models. The performance degrades but still outperform the layer-wise predictors.

Figure S1: (Left) Graph representation of a cell used in NAS-Bench-201 models, as defined by the authors, (Middle, Right) equivalent representation (with additional global node) and its adjacency matrix (without considering optimizations to remove "zero" and "skip-connect" operations) which are used in this paper.

Table S1: Training hyperparameters of the latency predictors.

| | |
|---|---|
| Batch size | 10 |
| Learning rate schedule | plateau (reduce learning rate by half if no improvement is seen for 10 epochs) |
| Initial learning rate | 0.0008 |
| Optimizer | AdamW |
| L2 weight decay | 0.0005 |
| Dropout ratio | 0.002 |
| Training epochs | 250 (early stopping patience of 35 epochs) |

Table S2: Performance of latency predictors of various devices on NAS-Bench-201: (i) D. CPU - Intel Core i7-7820X, (ii) D. GPU - NVIDIA GTX 1080 Ti, (iii) E. GPU - NVIDIA Jetson Nano, (iv) E. TPU - Google EdgeTPU, (v) M. GPU - Qualcomm Adreno 612 GPU, (vi) M. DSP - Qualcomm Hexagon 690 DSP.

| Error bounds | Accuracy [%] | | | | | |
|---|---|---|---|---|---|---|
| | D. CPU | D. GPU | E. GPU | E. TPU | M. GPU | M. DSP |
| **GCN** *(900 pts.)* | | | | | | |
| ±1% | 36.0±3.5 | 36.7±4.0 | 24.3±1.4 | 16.2±3.6 | 17.5±2.8 | 21.3±1.9 |
| ±5% | 85.2±1.8 | 85.9±1.9 | 82.5±1.5 | 64.0±5.7 | 67.5±7.4 | 77.5±2.6 |
| ±10% | 96.4±0.7 | 96.4±0.7 | 96.3±0.5 | 87.4±2.7 | 90.5±5.5 | 94.2±0.4 |
| **GCN** *(100 pts.)* | | | | | | |
| ±1% | 6.1±1.7 | 5.9±1.3 | 9.9±1.3 | 6.2±1.0 | 5.2±0.9 | 10.3±1.1 |
| ±5% | 27.9±5.5 | 28.7±3.6 | 44.6±4.0 | 30.0±3.6 | 24.9±3.4 | 48.0±3.8 |
| ±10% | 51.5±8.3 | 52.9±5.0 | 71.8±3.5 | 54.6±5.7 | 46.3±4.1 | 78.4±3.6 |
| **Layer-wise** *(900 pts.)* | | | | | | |
| ±1% | 3.5±0.2 | 4.2±0.2 | 6.1±0.3 | N/A | N/A | N/A |
| ±5% | 18.2±1.5 | 17.1±0.3 | 29.7±0.8 | N/A | N/A | N/A |
| ±10% | 29.6±1.1 | 32.6±1.2 | 54.0±0.8 | N/A | N/A | N/A |

## S1.3 Oracle NAS

In this section, we provide a detailed description of oracle NAS and the comparison between the layer-wise latency predictor-based oracle NAS and our GCN latency predictor-based oracle NAS (Figure S3). As noted in Section 3.1, in order to analyze the error introduced by an inaccurate latency estimation on NAS, we consider a

| (a) GCN - Desktop CPU | (c) GCN - Desktop GPU | (e) GCN - Embedded GPU |
| (b) Layer-wise - Desktop CPU | (d) Layer-wise - Desktop GPU | (f) Layer-wise - Embedded GPU |

Figure S2: Performance of latency predictors of various devices on NAS-Bench-201

set of experiments where a perfect searching algorithm, denoted by *oracle NAS*, is used to find the best possible model under a varying latency threshold. Here "perfect" means that the algorithm has the full knowledge about accuracy of all points and is always able to find the most accurate one[6], but its knowledge about latency of different models is potentially limited by how latency is estimated. For each latency threshold $l_{th}$, we begin by discarding all models which are believed to be too expensive according to the latency predictor in use. We then obtain the best model out of those which are left using our oracle search – this model is the *assumed best* but we are still not sure because it might have been falsely accepted due to imperfect latency estimation. Therefore we re-validate the assumed best model, this time using its measured latency, and only accept it if it truly falls below the latency limit. Otherwise we call it a *false positive* and discard it, repeating the aforementioned process with the second best point according to the initial search result. The first model encountered during this re-validation phase whose latency falls below the threshold is called the *effective best* for a given predictor with latency limit, and the effective best of a search when the ground-truth measured latency is always used is called *ground-truth best*.

As shown in Figure S3, we extensively study the difference between the assumed best and the effective best (introduced by false positives), as well as the difference between the accuracy of the ground-truth best and the effective best (introduced by false negatives) and some other accompanying metrics. Formally, for a set of models $\mathbb{S}$, a latency threshold $l_{th}$, latency predictor $\mathrm{pred}(\cdot)$ and measured latency $\mathrm{lat}(\cdot)$, we can define: (i) the set of false positives: $\{s \mid s \in \mathbb{S} \wedge \mathrm{pred}(s) < l_{th} \wedge \mathrm{lat}(s) > l_{th}\}$; (ii) the set of false negatives: $\mathrm{pred}(s) > l_{th} \wedge \mathrm{lat}(s) < l_{th}$; (iii) analogically true positives/negatives if comparison with $l_{th}$ is consistent between $\mathrm{pred}(\cdot)$ and $\mathrm{lat}(\cdot)$; (iv) and finally the set of truly positive points when $\mathrm{lat}(s) < l_{th}$.

Let us denote the set of: false/true negatives as $\mathbb{N}_f$ and $\mathbb{N}_t$ respectively, analogically false and true positives as $\mathbb{P}_f$ and $\mathbb{P}_t$, and the set of truly positives as $\mathbb{P}$. The *assumed best* is defined as: $s_f^\star = \arg\max_{s \in \mathbb{P}_f \cup \mathbb{P}_t} \mathrm{accuracy}(s)$; *effective best* as: $s_p^\star = \arg\max_{s \in \mathbb{P}_t} \mathrm{accuracy}(s)$; and *ground-truth best* as: $s^\star = \arg\max_{s \in \mathbb{P}} \mathrm{accuracy}(s)$. Then, Figure S3 shows the following metrics as functions of latency threshold:

- Top left: the number of *false positives* denotes how many models were considered below the limit incorrectly, i.e., $|\mathbb{P}_f|$.

- Top right: the number of *false negatives* denotes how many models were considered above the limit incorrectly, i.e., $|\mathbb{N}_f|$.

- Middle left: the *missed accuracy* denotes the accuracy difference between the ground-truth best model and the effective best model, i.e., $\mathrm{accuracy}(s^\star) - \mathrm{accuracy}(s_p^\star)$.

Figure S3: Oracle NAS results for desktop GPU, using GCN and Layer-wise latency predictors. Results are obtained on the NAS-Bench-201 dataset for desktop-GPU using both the layer-wise and GCN-based latency predictors. All latency thresholds are between 1-7ms with a step size of 0.1ms.

- Middle right: the related *latency prediction error* if a model was missed, i.e.,
$$\begin{cases} \mathrm{pred}(s^\star) - \mathrm{lat}(s^\star) & \text{if } s^\star \neq s_p^\star \\ 0 & \text{otherwise} \end{cases}$$

- Bottom left: the *over-claimed accuracy* denotes the accuracy difference between the assumed best model (i.e., including false positives) and the effective best model after removing false positives, i.e., $\mathrm{accuracy}(s_f^\star) - \mathrm{accuracy}(s_p^\star)$

- Bottom right: the related *latency prediction error* if a model was over-claimed, i.e.,
$$\begin{cases} \mathrm{pred}(s_f^\star) - \mathrm{lat}(s_f^\star) & \text{if } s_f^\star \neq s_p^\star \\ 0 & \text{otherwise} \end{cases}$$

# S2 Supplementary Material for Section 4: Accuracy prediction in NAS

## S2.1 Transfer learning from latency predictors to improve accuracy predictors

Latency predictors can improve the performance of accuracy predictor as shown in Table 2 of Section 3. The trained GCN captures the correlated features in the model which is useful to guide the training of a different GCN. Figure S4 further shows that the training process is improved. Y-axis is the percentage of models with predicted validation accuracy within the error bound relative to the actual validation accuracy. When initialized with the weight of latency/FLOPS predictor, the training process of accuracy predictor converges faster to better results.

In order to understand the underlying behavior and to improve the accuracy predictor proposed in Section 4.1, we plot the rankings produced by a standard predictor-based search (Figure S5 left) and by a predictor transferred from latency predictor against the ground-truth ranking (Figure S5 right). Even though the accuracy predictor with transferred knowledge performs better in predicting the accuracy values of the models overall, the gain in accuracy ranking (which is important to NAS performance) is not as much. This motivates BRP-NAS described in Section 4.

Figure S4: Training curves of the accuracy predictors without transfer learning (standard), with transfer learning from desktop GPU latency predictor (Init-GPU), and with transfer learning from FLOPS predictor (Init-FLOPS).

Figure S5: (Left) Rankings produced by a standard predictor-based search. (Right) ranking produced by a predictor transferred from a latency predictor, x-axis is the position of a model according to the ground-truth ranking using validation accuracy, y-axis represents the average position of model in a ranking produced by a relevant method and the dashed red line marks the $x = y$ diagonal (i.e., perfect ranking).

## S2.2 Motivation behind binary relation predictors

We propose the binary relation predictor in Section 4 based on the following observations and intuition.

**Observation 1: Ranking candidate models correctly according to their accuracy is more important than improving the absolute average accuracy of the accuracy predictor.**

When the number of training samples is to be minimized, like in NAS – the prediction quality of a GCN accuracy predictor can be improved by considering cheaper but roughly-correlated metrics, such as latency or FLOPS. However, even when using those cheaper metrics, the achievable prediction accuracy degrades significantly as the number of training samples becomes heavily limited as shown in Table 2 of Section 3. In order to maintain decent NAS quality even in those extreme cases, we propose to make more fundamental (compared to naively increasing a predictor's accuracy) changes in the predictor-based NAS by relaxing some of the current assumptions behind it.

More formally, we consider a predictor which gives each model a *score* rather than predicts the absolute accuracy a model. Let $P$ be the predicted order of models obtained from the estimated scores, i.e., $P_n$ is the model in the search space which achieves the $n$-th largest score as estimated by a predictor. Consequently, let $GT$ be the ground-truth order, i.e., $GT_n$ is the model which achieves the $n$-th highest validation accuracy. Furthermore, let $P(GT_n)$ be the position in $P$ of the $GT_n$ model, i.e., $P(GT_n) = m \iff P_m = GT_n$. Similarly, $GT(P_n)$ is

---

**Algorithm 1:** The proposed search method based on pairwise relation learning.

---

**Input:** *(i)* Search space $\mathbb{S}$, *(ii)* budget for predictor training $K$ (number of models), *(iii)* number of iterations $I$, *(iv)* latency limit $L_{max}$ and latency predictor $P_L$, *(v)* trade-off factor $\alpha$ between 0 and 1, *(vi)* overall maximum number of models we can afford to train $M$, $M > K$

**Output:** The best found model $m^*$

1  $m^* \leftarrow$ NONE
2  $\mathbb{C} \leftarrow \{\, s \mid s \in \mathbb{S} \wedge P_L(s) \leq L_{max} \,\}$  `// candidates, remove models predicted to be too expensive`
3  $\mathbb{T} \leftarrow \varnothing$  `// training set for the binary predictor`
4  $BP \leftarrow$ initialize binary predictor with weights from $P_L$  `// binary predictor, optional transfer from `$P_L$
5  **for** $i \leftarrow 1$ **to** *I* **do**
    &emsp;`/* Update training set for the predictor`
    &emsp;&emsp;`In each iteration we add `$K/I$` models in total */`
6  &emsp;$\mathbb{M} \leftarrow \{$ from $\mathbb{C}$, select the top $\alpha * K/I$ models and randomly select $(1-\alpha) * K/I$ models from the
    &emsp;top $|\mathbb{C}|/2^i$ which are not already in $\mathbb{T} \}$  `// this is completely random in i=1`
7  &emsp;**foreach** $m \in \mathbb{M}$ **do**  `// models higher in `$\mathbb{C}$` first`
8  &emsp;&emsp;$a \leftarrow$ train_and_validate($m$)  `// Get accuracy `$a$` of model `$m$
9  &emsp;&emsp;$\mathbb{T} \leftarrow \mathbb{T} \cup \{(m, a)\}$  `// Add the model-accuracy pair to the training set`
    &emsp;&emsp;`/* keep track of the trained models`
    &emsp;&emsp;&emsp;`throughout the entire procedure */`
10 &emsp;&emsp;**if** *latency*($m$) $\leq L_{max}$ **then**  `// check if `$m$` is truly below the lat. limit`
11 &emsp;&emsp;&emsp;update $m^*$ if $m$ happens to be better
12 &emsp;&emsp;**end**
13 &emsp;**end**
    &emsp;`/* Update the predictor */`
14 &emsp;**foreach** $\big((m_1, a_1), (m_2, a_2)\big) \in \mathbb{T}^2$ *s.t.* $m_1 \neq m_2$ **do**  `// possibly shuffle and batch`
15 &emsp;&emsp;$l \leftarrow$ softmax($BP(m_1, m_2)$)
16 &emsp;&emsp;$t \leftarrow$ softmax($[a_1, a_2]$)
17 &emsp;&emsp;optimize $BP$ to minimize KL-divergence between $t$ and $l$
18 &emsp;**end**
    &emsp;`/* Use the updated predictor to reevaluate the models in `$\mathbb{C}$` */`
19 &emsp;$\mathbb{C} \leftarrow$ sort $\mathbb{C}$ using $BP$ to compare models
20 **end**
    `/* At this point we stop training the predictor.`
    &emsp;`We have the final ordering of candidate models in `$\mathbb{C}$
    &emsp;`and we can still train `$M-K$` models to potentially find better `$m^*$` */`
21 $\mathbb{M} \leftarrow \{$ top $M - K$ models from $\mathbb{C}$ which have not been trained yet $\}$
22 **foreach** $m \in \mathbb{M}$ **do**  `// models higher in `$\mathbb{C}$` first`
23 &emsp;**if** *latency*($m$) $\leq L_{max}$ **then**  `// check if `$m$` is truly below the lat. limit`
24 &emsp;&emsp;$a \leftarrow$ train_and_validate($m$)
25 &emsp;&emsp;update $m^*$ if $m$ happens to be better
26 &emsp;**end**
27 **end**

---

the analogical reverse. It is easy to see that the performance of our predictor-based NAS should be maximized when $P = GT$, regardless of the values predicted by the scoring function. Although a perfect accuracy predictor (in the absolute sense) would produce the perfect ordering of models, we argue that learning the perfect accuracy function is more challenging than learning a function which is only supposed to produce faithful ordering of models.

**Observation 2: Learning a binary relation rather than predicting absolute models.**

Taking a step further, the predicted order $P$ even need not be produced by a scoring function. Instead, we lean on the fact that any antisymmetric, transitive and connex *binary relation* produces a linear ordering of its domain. Thus, NAS could be solved by learning a binary relation rather than predicting absolute accuracy values. This is a very important observation to maximize sample efficiency, since the reformulated binary relation changes the number of training samples for the predictor in a function of trained models to $O(n^2)$, rather than $O(n)$ in the standard approach. This provides the predictor with more opportunities to learn efficiently when $n$ is limited.

We quantify the quality of different rankings $P$ produced by the proposed binary relation predictor together with different variations of the standard predictor. All predictors were trained for 200 times, each time using a randomly sampled set of 100 models. Then they are used to sort all 15k models in the NAS-Bench-201 dataset to produce a ranking. We compared the predicted rankings by considering their correlation to the GT rankings in Figure 5 (middle) of Section 4. The average Spearman-$\rho$ correlation coefficient between the position in

prediction ranking and that in GT ranking shows that the proposed binary relation predictor achieves the best ranking correlation out of all our experiments. However, despite producing the best results globally, the binary predictor did not yield the best NAS results. This had led us to the next observation.

**Observation 3. Top-K rankings are important.**

Even though $P = GT$ maximizes the performance of predictor-based NAS, achieving perfect correlation between the two rankings is very challenging in practice considering a limited number of training samples. Although errors are expected to occur *somewhere* in the predicted ranking, in the context of NAS it is especially important to make sure that those errors are minimized in the top of the rankings, otherwise even a very well correlated ranking might fall short to a less optimal alternative.

When closely examining the results obtained by running the binary relation predictor, we saw that even though the global correlation was very good, the best performing models happened to be burdened with a relatively higher error than the rest of the search space, as indicated by the red circle in Figure 5. In the context of NAS, any ranking $P'$ which satisfies $P'_1 = GT_1$ is as good as the perfectly correlated ranking $P = GT$, and analogically, any ranking $P''$ for which $GT(P''_1)$ is very high is less likely to yield good results in practice, regardless of their global landscapes.

Section 4 has introduced BRP-NAS - a NAS method based on binary relation predictor combined with an iterative data selection strategy. Algorithm 1 describes the steps to search for the best model based on pairwise relation learning and to find better models by focusing on high performing models. Figure S6 shows that the proposed method achieves the best NAS performance. The iterative training approach helps with the top model performance even though achieves worse results globally.

### S2.3 BRP-NAS details

We train each predictor using the hyperparameters listed in Table S3. A technical detail related to accuracy predictors is that when training them we used fixed accuracy values for each model by considering only the seed 888 from NAS-Bench-201, and by taking an average accuracy from NAS-Bench-101 (for DARTS we were taking accuracy after a single training, whatever that was). Even though the predictors were trained using noise-free data, when performing a search and reporting results we were returning a random accuracy for each model, thus resulting in a potential distribution shift between training and testing sets for our predictors, induced by the stochastic nature of the training process. More details about the impact of SGD, and related experiments, can be found in Section S3.2.

Table S3: Training hyperparameters of the accuracy predictors.

|  |  | K=100 | K=50 | K=25 |
|---|---|---|---|---|
| batch size | normal | 50 | 32 | 16 |
|  | binary | 64 | 32 | 32 |
| Learning rate schedule | cosine annealing |  |  |  |
| Initial learning rate | 0.00035 |  |  |  |
| Optimizer | AdamW |  |  |  |
| L2 weight decay | 0.0005 |  |  |  |
| Dropout ratio | 0.2 |  |  |  |
| Training epochs | 250 (early stopping patience of 35 epochs) |  |  |  |

For all NAS experiments, the training set for predictors was discovered online (i.e., no prior knowledge was assumed), therefore, we did not have access to a separate validation set – whenever a validation set would have been used, we used training set instead.

To perform NAS with our predictor, in the first phase, we use a randomly selected small set of $K$ models from the search space to train the predictor (with or without iterative data selection). Then in the second phase, we use the predictor to score all the models in the search space in order to find potentially the most efficient one. The predictor could be trained in the second phase if required, but we did not do that in order to check the achievable performance when $K$ is upper-bounded.

When training the predictor iteratively, we always use 5 iterations. It means that different number of models ($K/5$) were trained and added to the predictor's training set in each iteration.

#### S2.3.1 Ablation studies

Figure S6 summarizes the results of NAS using various GCN-based predictors, and Figure S7 additionally shows results with different total number of models used to train the predictors. BRP-NAS, which utilizes binary

relation predictor trained via iterative data selection, has the best performance comparing to other approaches, and is able to achieve competitive results even when very few models are used to train it.

Figure S8 additionally shows the effects of using different activation functions and related labels. Specifically, we tried replacing our proposed 2-way softmax with analogical 1-way sigmoid activation (which some people can consider more canonical when dealing with probability of opposite events) with either soft or hard labels. Soft labels were obtained by by linearly interpolating between accuracies of the related networks $(a, b)$: $l_{a,b} = \frac{\mathrm{acc}(a) - \mathrm{acc}(b) + 1}{2}$, whereas hard labels were simply 1 if model $a$ is more accurate or 0 otherwise. In both cases the predictor was trained to minimize binary cross-entropy loss. As can be seen, there is no visible difference between softmax and sigmoid activations when both use soft labels, however, switching to hard labels results in a reduced sample efficiency. We decided to stick with the softmax activation as it can be easily extended to the $n$-ary case (although doing so is outside the scope of this paper) and thus we consider it more general.

Figure S6: (Left) Comparison of NAS performance with the standard GCN predictor, GCN predictor with transfer learning (from desktop GPU, Embedded TPU and FLOPS predictors) and BRP-NAS, all trained non-iteratively. (Middle, Right) Comparison with predictors trained via iterative approach with $\alpha = 0$ (middle) and $\alpha = 0.5$ (right).

Figure S7: Comparison of GCN accuracy predictors and our BRP-NAS predictor (with $\alpha = 0.5$) under different total number of models used to train the predictor: 100, 50 and 25, respectively. In all cases, 5 iterations were used.

# S3 Supplementary Material for Section 5: End-to-end results

## S3.1 Details of the baseline NAS algorithms used in the paper

**Aging Evolution.** We run aging evolution with pool size 64 and sample size 16, values which we found work well for NAS-Bench-201 models.

**REINFORCE.** Similar to [1], we use a single cell LSTM controller which is trained with REINFORCE (no PPO).

**Random search.** We select models randomly by picking 6 numbers from the range of 1-5 uniformly.

## S3.2 SGD-induced randomness in NAS-Bench-201

As mentioned in the main paper, when comparing different algorithms on NAS-Bench-201 we observed that eventually AE surpasses our BRP-NAS due to its ability to train models multiple times. To validate our reasoning and at the same time formally measure the effect of SGD randomness on NAS, we conducted a set of experiments in addition to the ones presented in the main text. Specifically, we checked: *(a)* performance of both BRP-NAS and AE when SGD noise is not present (using only seed 888), and *(b)* running a modified AE where we remember trained models and avoid mutations that would result in the same model being trained more than once (cached

Figure S8: Effects of using a different activation functions with our binary accuracy predictor.

Figure S9: Effects of the SGD-induced randomness on NAS performance. Runs labeled with "(seed 888)" were using only one seed from NAS-Bench-201. Aging Evolution (cached) was not allowed to train the same model multiple times.

mode). The results are summarized in Figure S9. We can see that indeed AE surpasses our predictor due to its ability to train models multiple times and that the gap is proportional to the level of the SGD-induced noise. At the same time we can see that the difference in results obtainable with and without considering the SGD noise is surprisingly high – the performance of the best model increases, in terms of the top-1 accuracy, by almost 2% absolute. Interestingly, we did not observe a similar gap when running analogical experiments in a constrained setting (5ms, D-GPU), suggesting that the huge gap observed for the top performing models is not necessarily common, even within the same search space.

## S3.3 Comparison to Prior Work on NAS-Bench-101

We compare BRP-NAS with the previously published predictor-related NAS on NAS-Bench-101. We first train the predictor using the validation accuracy from 100 models, then we train the subsequent models which are picked from the top-ranked models returned by the predictor. As shown in Figure S10, BRP-NAS finds a model with higher final test accuracy on CIFAR-10 dataset (94.22%) using fewer steps (140 trained models) than the work under comparison.

Figure S10: Comparison to prior work on NAS-Bench-101 dataset.

## S3.4 DARTS search space

We run BRP-NAS on the DARTS search space, as described in Section 5. All models were trained using the official DARTS implementation[7] to ensure fair comparison. When quantifying searching cost, we assumed one model takes 1 GPU day to train, which we empirically observed to be true on average when training our models using V100 GPUs.

The results, additionally to Table 4 presented in the main paper, are shown in Figure S11. Our first observation was that the search space is actually quite dense with good models and – to our surprise – our simple random search algorithm (take a random model and train) already was able to achieve very good results – comparable with differentiable search for the same searching budget. This is also shown visible in good performance of our BRP-NAS during the first iteration. Because of that, we also include more detailed comparison against random search – to validate if our approach can actually improve. From our results, we can see that indeed as we move onto to the second and third iterations of our searching algorithm, the gap between random search and ours increases, suggesting that the predictor is able to extract useful features and use them to identify better models. On the other hand, the gap between random search and ours in the first iteration (first 20 models, should be close to each other) suggests that the results can still be influenced by the random nature of the first few selected models – unfortunately, due to limited computational resources we weren't able to run each algorithm for more than 3 times (each time training up to 60 models, as presented in the figure). We suspect that the gap between random search and BRP-NAS should remain if we take average of more runs, but to test it robustly one would probably need to do more exhaustive evaluation.

Figure S11: Performance on DARTS search space.

## S4   Supplementary Material for Section 6: Latency prediction benchmark

We run each model in NAS-Bench-201 on the follow devices and run-times. (i) Desktop CPU - Intel Core i7-7820X - TensorFlow 1.15.0, (ii) Desktop GPU - NVIDIA GTX 1080 Ti - TensorFlow 1.15.0, (iii) Embedded GPU - NVIDIA Jetson Nano - TensorFlow 1.15.0, (iv) Embedded TPU - Google EdgeTPU - TensorFlow Lite Runtime 2.1.0, (v) Mobile GPU - Qualcomm Adreno 612 GPU - SNPE 1.36.0.746, (vi) Mobile DSP - Qualcomm Hexagon 690 DSP - SNPE 1.36.0.746.

In Figure S12, we show the scatter plots of models taken from NAS-Bench-201 dataset that illustrates the connection between the latency of various devices and number of parameters/FLOPS. Each point in the plots represents the average latency of running a model on the stated device. We can see that latency is not strongly correlated with FLOPS or number of parameters. These metrics are unreliable proxies to predict latency.

Figure S13 and Table S4 illustrate the latency correlation between devices. Most of the metrics are not strongly correlated which indicates that having a dedicated latency predictor trained for each class of devices is necessary to provide good latency estimation. This motivates us to provide LatBench as a latency dataset.

Table S4: Latency correlation between various devices.

|        | D. CPU | D. GPU | E. GPU | E. TPU | M. GPU | M. DSP |
|--------|--------|--------|--------|--------|--------|--------|
| D. CPU | 1.000  | 0.997  | 0.700  | 0.844  | 0.751  | 0.727  |
| D. GPU | 0.997  | 1.000  | 0.702  | 0.844  | 0.752  | 0.728  |
| E. GPU | 0.700  | 0.702  | 1.000  | 0.574  | 0.866  | 0.821  |
| E. TPU | 0.844  | 0.844  | 0.574  | 1.000  | 0.548  | 0.690  |
| M. GPU | 0.751  | 0.752  | 0.866  | 0.548  | 1.000  | 0.821  |
| M. DSP | 0.727  | 0.728  | 0.821  | 0.690  | 0.821  | 1.000  |

Figure S12: In most cases, FLOPS and the number of parameters are not a good approximation towards run-time latency on-device.

(a) Desktop CPU vs Desktop GPU

(f) Desktop GPU vs Mobile GPU

(k) Embed. TPU vs Mobile GPU

(b) Desktop CPU vs Embed. GPU

(g) Desktop GPU vs Mobile DSP

(l) Embed. TPU vs Mobile DSP

(c) Desktop CPU vs Mobile GPU

(h) Embed. TPU vs Desktop CPU

(m) Embed. GPU vs Mobile GPU

(d) Desktop CPU vs Mobile DSP

(i) Embed. TPU vs Desktop GPU

(n) Embed. GPU vs Mobile DSP

(e) Desktop GPU vs Embed. GPU

(j) Embed. TPU vs Embed. GPU

(o) Mobile DSP vs Mobile GPU

Figure S13: Latency differs for each class of devices.

## Footnotes

[3]One of the reasons behind this decision was to have a unified network representation consistent between NAS-Bench-101 and 201.

[4]For skip-connections, before detaching we make sure that all direct predecessors of the node are instead connected directly to all direct successors of the node.

[5]Since the optimized nodes are actually still present in the graph (but detached from anything else) we simply considered them "typeless" and assign zeros-only vectors to them.

[6]To simplify this process, we did not consider the SGD noise and instead only used accuracy values from NAS-Bench-201 with seed 888.

[7]https://github.com/quark0/darts