[Reviews · NeurIPS 2020]

Review 1

Summary and Contributions: The paper aims to improve the sample-efficiency of neural architecture search with latency constraint. Its contribution is three-folded: 1) It introduces a latency prediction model and shows that its weights can be used for an accuracy predictor which improves its accuracy. 2) It introduces BRP-NAS uses a binary accuracy predictor and an iterative data selection strategy. 3) It provides an extension to NASBench201, called LatBatch, which contains the latency of the models on a variety of hardware. The contributions of this paper are quite outstanding. However, there are some weaknesses regarding the some technicle details (explained below), so I reduce the score. I hope the authors can address the problems and make it a solid paper.

Strengths: - All proposed methods are sound and convincing. - The empirical results are strong and support their arguments. - The BRP-NAS approach looks simple but effective. - There are several thought-provoking observations including the ones in the supplementary materials. - Hopefully, the proposed LatBench may shift the focus of the community from FLOPS or model size to latency, a more realistic metric for model deployment.

Weaknesses: In short, the paper misses some critical details and some explanations are not clear. After reading the paper and the supplementary materials, I still believe it is hard to reproduce their results. - The outputs of the binary relation predictor model shown in Figure 4 are not symmetric. Let F be the model and m1 and m2 be two model archietectures, (p1, p2) = F(m1, m2) and (p2', p1') = F(m1, m2). It is clear that p1 != p1' and p2 != p2' since the model just concatenates two graph embedded vectors and feeds it to a linear classifier. I wonder how the authors handle this problem. - The meaning of the shaded regions of Figure 2 (right), Figure 5 (left), and Figure 6 (left) is never explained in the caption or main text. Morevoer, it neighor represents standard deviation or the range between the best and worst case because the line with the same color can go outside the shaded region. For example, the solid red line in Figure 5 below the red shaded region between 60 and 90 trained models. - In the equation after line 156: "\sigma ( A H^l W H^l)" should be "\sigma( A H^l W )". Is this a typo? Otherwise, this is not a graph convolution layer. I have low tolerance with a typo in the "only one" equation in a paper. - It is not clear how the adjacency matrix A is defined in line 155. A_ij is the edge of node i to node j or node j to node i? Although I can figure it out from Figure S1 in supplementary materials, it is not clear in the main paper. - Following the previously point, based on Figure S1, the graph convolution operation propogates the features "backward" in the graph (in the opposite direction of an edge). I wonder if the authors can explain why you made this decision instead of adding edges for each node to the "global" node and following the edge direction to do graph convolution? - The original GCN (Kipf and Welling) uses normalized adajancy matrix with renormalization trick. However, this paper uses the original adjacency matrix without explantation or ablation study. I wonder if there is a reason behind not using the conventional graph convoltuional operation.

Correctness: In general, the experical methodology is correct.

Clarity: The paper is well written in terms its high-level structure. However, some technical details can be confusing (see weakness section). There are also so some typos: - line 140: "Can discover Pareto-optimal..." --> missing subject, do you mean "Can we discover Pareto-optimal..." ? - Figure 2 (right): in the legend, "Agining" --> "Aging" - line 178: "in in" --> "in"

Relation to Prior Work: Yes, the mothods are quite novel and the differences are quite explicit.

Reproducibility: No

Additional Feedback: - Recently, it is quite common to use half-precision floating-points (FP16) on modern GPUs or use INT8 quantization on CPUs to improve the latency. It would great if the authors can include these configurations in their LatBench. - Regarding the binary relation predictor, it is more common to have one output with sigmoid rather using a 2-way softmax classifier. I wonder if there are some reason behind choosing this instead of a more common and more efficient way.


Review 2

Summary and Contributions: 1. Latency prediction, propose to use a GCN module to predict the latency. 2. Binary relative accuracy predictor, using a GCN module to learn the binary relation of relative accuracy between two neural network architectures. 3. Prediction-based NAS, NAS with the binary relative accuracy predictor 4. latency benchmark: LatBench is the first large-scale latency measurement dataset for NAS under multiple hardware (desktop CPU/GPU, embedded GPU/TPU, mobile GPU/NPU), based on the dataset NASBench-201.

Strengths: 1. Utilize GCN to model the architectures. The computational graph could be represented as a GCN directly. 2. Focus on the accuracy ranking of neural networks. 3. Introduce a latency benchmark, which provides the latency of each model in NASBench-201 on a wide range of devices.

Weaknesses: - This paper missed some previous related works. For example, the work "Multi-objective neural architecture search via predictive network performance optimization" also proposed to use GCN for architecture performance prediction (see https://openreview.net/forum?id=rJgffkSFPS). Therefore, the idea of using GCN based prediction is not novel. Pls discuss the key differences. - Besides the Benchmarks, pls add experiments on real search space, such as DARTS space or MobileNetV3 space. The benchmark is very small, and easy to overfit. For example, if you direct select the large FLOPs models in Benchmarks, the performance is very competitive, surpassing many SOTAs. Add experiments on ImageNet. - Move the details of latency prediction model and binary relative accuracy predictor from supplementary material to main manuscript, because these two parts are important for the work. - No explanation how to address the problem when the cycle exists in binary relative accuracy predictor. For example, acc(m1) > acc(m2), acc(m2) > acc(m3), acc(m3) > acc(m1)

Correctness: In the supplementary material, the part of GCN predictor is correct. However, the proposed binary relation predictors doesn't avoid the cycle among ranking relation.

Clarity: Overall, the writing is good, but missing the most recent related work.

Relation to Prior Work: Missing references: Multi-objective neural architecture search via predictive network performance optimization D-vae: A variational autoencoder for directed acyclic graphs Discuss the differences with there two papers, which are also based upon GCN.

Reproducibility: No

Additional Feedback: The study on the binary relative accuracy and the latency prediction model is valuable. =========Post Rebuttal============= After reading the rebuttal, I lean to accept this paper. Pls revise the paper, the reviewer would like to see the final version.


Review 3

Summary and Contributions: The authors investigate the problem of neural architecture search with a focus on latency prediction. Their primary contributions are: a latency predictor using graph convolutional networks, a new strategy for architecture search using binary comparisons, and an extension of NAS-Bench-201 with latency computations on various hardware targets. In terms of latency prediction, the authors show the weakness of using FLOPS as well as layer-wise prediction and instead opt for an end-to-end graph convolutional network predictor. Results show that the predicted latency more closely matches the actual measured latency than layerwise predictions. The work also presents BRP-NAS (binary relation prediction), which trains a forked GCN-based network to predict the pairwise rankings. When combined with an iterative training approach, the method beats aging evolution and other reported results. To facilitate further research, the authors also release LatBench, which is an extension of NAS-Bench-201 which measures the real-world latency on various hardware after removing unused nodes.

Strengths: * Clearly shows the weaknesses of existing latency predictors (FLOPS, layer-wise) and presents a predictor-based alternative that shows high correlation with the measured latency. The benefit of this predictor is obvious when tested with a fixed oracle NAS algorithm against the layerwise alternative. * While the idea of using GCN for architecture search isn't completely novel, the choice to use GCN for latency prediction seems novel. The results from this paper may motivate future exploration with GCN in the NAS field. * The authors actively make efforts to improve the reproducibility of the NAS field by releasing LatBench, which improves upon NAS-Bench-201.

Weaknesses: * The GCN-based predictor and experiments don't have open-sourced code (not mentioned in the main paper or supplement), however the authors do provide detailed descriptions. * Some correctness issues (see next section) * The paper presents 2 important NAS objectives: latency optimization and accuracy optimization. However, the BRP-NAS (section 4) seems out-of-place since the rest of the paper deals with latency prediction. It nearly feels like BRP-NAS could be a separate paper, or Section 3 was used only to suggest using GCN (in this case, why not directly start with accuracy prediction with GCN?). * The analysis on BRP-NAS is also somewhat barebones: it only compares against 3 basic alternatives and ignores some other NAS (e.g. super-net/one-shot approaches, etc...). * Unclear if code will be released, as the GCN implementation may be hard to reproduce without original code (though the author's descriptions are fairly detailed and there is more information in the supplement).

Correctness: The empirical methodology mostly seems correct: the authors are careful to report the results of multiple randomized runs for some experiments (Figure 2 left and Table 3 are notably missing this error analysis). The main concern for correctness is that in Section 3 (Figure 2) is that the comparison of layerwise latency might not be fair as the layerwise metrics don't have access to the true latency and many selected models may be disqualified (the GCN predictor has access to 900 training models that the layerwise did not have access to).

Clarity: Paper is mostly well written except that there seems to be a gap in focus as Section 3, 4, 5 swap between discussing latency, then accuracy, then latency again. The tables, figures, and diagrams in the paper are well-made and enhance the understanding of the work.

Relation to Prior Work: Related work section acknowledges the past work in NAS accuracy and latency prediction. While the authors' work appears mostly novel in the combination of GCN and latency prediction, some acknowledgement of prior GCN work in NAS could be valuable.

Reproducibility: Yes

Additional Feedback: * [page 4, Figure 2] The experiments use a strict latency constraint, but a layerwise predictor does not have access to the end-to-end latency. Do the architectures discovered by the layerwise predictor that exceed the constraint get thrown out? If so, isn't this an unfair comparison of the two approaches (you consider fewer valid models with layerwise)? * [page 7, Figure 5] Why does the BRP with iterative training have lower Spearman's rho than the middle plot? * [page 7, Table 3] Is this the average/median of many runs? NAS is very sensitive to randomness and it is possible that the advantage of BRP-NAS is insignificant considering noise. Some clarification on how the "100 models" and "top 45 models" are selected could be useful as well. * [page 8, Figure 6] It would be more fair for comparing BRP-NAS with AE if you used the same latency predictor. Also, the right/left order of the caption is confusing. * It should not be very surprising that a end-to-end predictor performs better than a layerwise predictor when evaluated on a fixed macro-architecture (i.e. number of layers/cell). Does the GCN also generalize better than layerwise prediction when using an unseen macro-architecture (e.g. evaluated using a rank-based metric)? * Extending the previous point: do the approaches presented extend to larger cell sizes than the ones observed? Realistically, NAS-Bench-101 and 201's cells are smaller than an actual search space that someone would use for SOTA research. Is there any evidence that your methods generalize better to larger spaces?


Review 4

Summary and Contributions: This paper proposes a graph based approach to predict the performance of NAS on energy efficient devices. To achieve that the authors used a GCN to predict some of the benchmarks that are unique in the mobile device setting such as latency.

Strengths: The paper is clearly motivated to use GCN to encode architectures. A prediction on the performance of mobile devices is also one of the key concerns on NAS.

Weaknesses: The proposed approach can be used towards a wider range of application other than hardware-aware NAS. For example, the authors could have used the method to improve the performance estimations of NAS even without the hardware constraints.

Correctness: Authors has done a wide range of experiments to justify the effectiveness of the graph based approach for performance prediction.

Clarity: Paper is well written with clear demonstrations and experiments.

Relation to Prior Work: Prior work is sufficient for this work as it compared multiple NAS frameworks.

Reproducibility: Yes

Additional Feedback:

[Author Response · NeurIPS 2020]

We thank the reviewers for their insightful comments – we will make sure that all of them are adequately addressed in
the revised paper, including: missing references, relevant ablations, more NAS results, more latency measurements,
discussion about limitations and design choices – we wholeheartedly agree that these will significantly strengthen the
paper. *We will release all the source code and data.* Meanwhile, please find our detailed responses below.

**Novelty and contribution:** Both latency and accuracy are studied in this paper as they are critical for hardware-aware
NAS. We show how improvements in predicting either of them impacts the outcome of the search, and how imprecise
prediction of one metric can limit potential gains from improving the other – something that has not been systematically
studied before in the context of NAS. Building on top of that, our second contribution is to show an efficient way of
improving estimations of both accuracy and latency by using two versions of our GCN-based predictor, tailored for their
respective metrics, resulting in a new SOTA on common NAS benchmarks. We want to emphasize that even though
accuracy and latency prediction can be studied independently, they are both important in order to improve HW-aware
NAS – we will improve writing in order to make the connection clearer (**R3**). ● *Key differences from Shi et al. and*
*D-VAE (R2):* There are substantial differences between our work and Shi et al. The key idea of Shi et al. is predicting
accuracy with a GCN. On the contrary, our work covers latency prediction, binary relation prediction (for accuracy),
and combining both for multi-objective NAS. Additionally, our work on the binary relation learning and iterative data
selection is a novel solution to the open problem raised by Shi et al. (but was not addressed in their work) that the
ranking of models is more important than their absolute accuracy. D-VAE is complementary to our work as we could
use D-VAE in place of GCN; studying whether D-VAE further improves the results is an interesting open problem.

**Relation learning – symmetry and cycles:** We use the binary predictor as a drop-in replacement for the comparison
operator in the standard Python sorting function. In the case of symmetric and/or non-transitive relation, the final order
of elements will depend on their initial order and the implementation of the sorting algorithm. ● *Symmetry (R1):* (To
avoid confusion we will talk about anti-symmetry *of the relation* which is related to symmetry of the predictor). To
encourage anti-symmetry, we include both pairs of model architectures, i.e. (m1,m2) and (m2,m1), in the training
set and verify that the resulting relation is highly anti-symmetric, i.e., (p1-p2)(p1'-p2')>0 with probability 0.98, as
measured on 1000 randomly sampled points from NAS-Bench-201. ● *Cycles (R2):* Based on the comments, we ran
an experiment to identify cycles and found that for a sample of 1000 models there are more than 10 million cycles
(but only 31 for 300 models), suggesting that the number of cycles can grow exponentially. Even though we handle
cycles randomly, we are not concerned with them due to our strong empirical results. However, we do agree that fully
exploring their impact is an important research question.

**Bigger search space / unseen macro architecture:** Regarding search space size and scalability: even though we
primarily evaluated our approach on NAS-Bench-201, it performed very well when ported to NAS-Bench-101 (which
is an order of magnitude larger). Following the suggestion of **R2**, we are currently evaluating our methodology on the
DARTS search space and will include the results in the final paper.

**Layer-wise latency predictor is also trained with end-to-end latency:** The layer-wise predictor is calibrated with
the end-to-end latency by a scaling factor, i.e. exactly the same number of training examples – *(model, end-to-end*
*latency)* pairs – are used to train both predictors, which addresses (**R3**)'s concern on fairness. We will clarify the
training part in the revision. In latency-constrained NAS, the architectures discovered by any predictor that exceed the
constraint are discarded for a fair comparison (**R3**).

**GCN design decisions, typos, clarifications (R1):** We ran ablation studies based on the comments. ● *Normalizing*
*adjacency matrix:* We have tried different normalizations and the best result is achieved without normalization. *Softmax*
*and sigmoid* activations lead to similar results. *Global node and flow direction:* Both forward propagation (with a global
sink) and backward propagation (with a global source) result in similar performance. ● *LatBench with quantization:*
Thanks for the great suggestion. We already support INT8 models on EdgeTPU and Snapdragon DSP, and will release
LatBench with FP32/FP16/INT8 on supported platforms by November. ● *Typos, clarifications:* We will correct all typos
(especially, $\sigma(AH^lW)$) and add details on every term (e.g., $A_{ij}$). Shaded regions in figures mark interquartile range.

**Comparison to other NAS, other questions (R3):** ● We use Aging Evolution (AE) as the major baseline as it is
shown to perform best in the NAS-Bench-201 paper. We are up to 3x more sample efficient than the current best
(line 237 and Fig. 6 in the paper). Also, we beat all other SOTA methods on NAS-Bench-101. ● Fig. 6 compares
BRP-NAS with (AE+layer-wise) as AE and layer-wise are SOTA for NAS and latency predictions, respectively. Indeed
the performance of AE is improved by our predictor, which is highlighted in Fig. 2. ● Our result in Table 3 is averaged
from 32 runs. ● The BRP with iterative training has a lower Spearman-$\rho$ than the plain BRP as we are not concerned
about the ranking of low performing models and focused on high performing ones at the expense of global ranking
quality (see line 242-245 in the paper, and S2.1 Observation 3 in the SM).

**Applying our approach to NAS w/o hardware constraints (R4):** We already have results on this in Section 4. Fig. 6
(left) and Table 3 show exactly that BRP-NAS outperforms SOTA in unconstrained settings, e.g. accuracy prediction of
BRP-NAS vs AE. Generalization of our approach to broader settings (eg. energy or memory usage aware optimization)
is a fascinating future direction of research.

[Meta-Review · NeurIPS 2020]

The paper analysis of existing latency predictors is solid and the proposed method seems to provide a real improvement to the task of predicting latency. The overall performance lift is not demonstrated in a fully convincing manner and there could be some room for more extensive experiments. Still, all things considered, a solid method for predicting latency is relevant to NAS making this paper interesting to the NeurIPS community